# Comparative Metagenomics Highlight a Widespread Pathway Involved in Catabolism of Phosphonates in Marine and Terrestrial Serpentinizing Ecosystems

Eléonore Frouin,[a] Aurélien Lecoeuvre,[a] Fabrice Armougom,[a] Matthew O. Schrenk,[b] Gaël Erauso[a]

[a]Aix-Marseille Université, Université de Toulon, CNRS, IRD, MIO UM 110, Marseille, France
[b]Department of Microbiology and Molecular Genetics, Michigan State University, East Lansing, Michigan, USA

Eléonore Frouin and Aurélien Lecoeuvre contributed equally to this work. Author order was determined by anteriority in this study.

**ABSTRACT** Serpentinizing hydrothermal systems result from water circulating into the subsurface and interacting with mantle-derived rocks notably near mid-ocean ridges or continental ophiolites. Serpentinization and associated reactions produce alkaline fluids enriched in molecular hydrogen, methane, and small organic molecules that are assumed to feed microbial inhabitants. In this study, we explored the relationships linking serpentinization to associated microbial communities by comparative metagenomics of serpentinite-hosted systems, basalt-hosted vents, and hot springs. The shallow Prony bay hydrothermal field (PBHF) microbiome appeared to be more related to those of ophiolitic sites than to the Lost City hydrothermal field (LCHF) microbiome, probably because of the meteoric origin of its fluid, like terrestrial alkaline springs. This study emphasized the ubiquitous importance of a set of genes involved in the catabolism of phosphonates and highly enriched in all serpentinizing sites compared to other ecosystems. Because most of the serpentinizing systems are depleted in inorganic phosphate, the abundance of genes involved in the carbon-phosphorus lyase pathway suggests that the phosphonates constitute a source of phosphorus in these ecosystems. Additionally, hydrocarbons such as methane, released upon phosphonate catabolism, may contribute to the overall budget of organic molecules in serpentinizing systems.

**IMPORTANCE** This first comparative metagenomic study of serpentinite-hosted environments provides an objective framework to understand the functioning of these peculiar ecosystems. We showed a taxonomic similarity between the PBHF and other terrestrial serpentinite-hosted ecosystems. At the same time, the LCHF microbial community was closer to deep basalt-hosted hydrothermal fields than continental ophiolites, despite the influence of serpentinization. This study revealed shared functional capabilities among serpentinite-hosted ecosystems in response to environmental stress, the metabolism of abundant dihydrogen, and the metabolism of phosphorus. Our results are consistent with the generalized view of serpentinite environments but provide deeper insight into the array of factors that may control microbial activities in these ecosystems. Moreover, we show that metabolism of phosphonate is widespread among alkaline serpentinizing systems and could play a crucial role in phosphorus and methane biogeochemical cycles. This study opens a new line of investigation of the metabolism of reduced phosphorus compounds in serpentinizing environments.

**KEYWORDS** serpentinization, phosphonate catabolism, comparative metagenomics, hydrothermal systems

Serpentinite-hosted hydrothermal systems are the product of the hydration of ultramafic rocks (i.e., peridotite) originating in Earth's mantle. The serpentinization process produces serpentine minerals and emits hyperalkaline fluids enriched in hydrogen gas, methane, and

Address correspondence to Gaël Erauso, gael.erauso@mio.osupytheas.fr, or Fabrice Armougom, fabrice.armougom@mio.osupytheas.fr.

The authors declare no conflict of interest.

small organic molecules (e.g., formate, acetate, and methanol) (1, 2). These products of the serpentinization and associated reactions are assumed to feed the microbial inhabitants of these ecosystems (3). In submarine environments, discharges of highly reduced alkaline fluids precipitate upon mixing with surrounding seawater, forming large edifices of brucite and carbonate (4). The mixing of hydrothermal fluids with seawater also sustains ionic and redox gradients within the porous chimney walls. Similar gradients are thought to have played a role in the development of metabolic processes during the early evolution of life on Earth (5) and perhaps other planets (2). The emblematic Lost City hydrothermal field (LCHF), located at ~800 m below sea level (mbsl), 15 km off the Mid-Atlantic Ridge, has been the only submarine system of this type studied for many years and still represents a reference model for serpentinization-influenced geochemical processes and associated microbial ecosystems (6, 7). However, the sampling of deep-sea serpentinizing systems is technically challenging, and therefore, other, more accessible serpentinizing environments have been investigated, such as continental ophiolites, portions of ancient seafloor obducted onto continental margins (3). The Prony Bay hydrothermal field (PBHF), recently rediscovered at a shallow depth (<50 mbsl, bay of Prony, New Caledonia [8]), represents an alternate example of a marine serpentinizing ecosystem. There, like at LCHF, tall brucite-carbonate chimneys vent hyperalkaline fluids enriched in $H_2$ and $CH_4$, at lower temperatures (maximum, 42°C) than reported at LCHF (up to 90°C) (9). Because of their distinctive geologic settings (mid-ocean ridge versus suprasubduction zone for LCHF and PBHF, respectively), the two hydrothermal systems differ in their circulating fluids. Indeed, PBHF is alimented by meteoric waters (instead of seawater as at LCHF) percolating through the thick peridotite nappe (>1.5 km) and discharges low-salinity fluids into the lagoon (10). Thus, from a hydrogeological point of view, PBHF appears to be an intermediate between marine and terrestrial serpentinizing systems (10). This peculiar situation was also confirmed at a microbial community level, since PBHF shared several phylotypes with both continental and submarine serpentinizing ecosystems, such as members of the genus *Hydrogenophaga*-"*Serpentinomonas*," typical of terrestrial systems, or two distinct phylotypes of uncultivated *Methanosarcinales* (9–11), first discovered at LCHF (7) and The Cedars (12). Besides these few phylotypes, microbial communities inhabiting the serpentinizing ecosystems showed limited similarity (13, 14). This observation suggests that unknown environmental factors influence the taxonomic distribution beyond the physicochemical conditions imposed by serpentinization. In natural ecosystems, many different species possess similar metabolic capabilities and can potentially play the same ecological roles depending on the biogeography (15). Therefore, differences in the microbiome compositions of serpentinizing systems from distinct geographic sites do not necessarily reflect different functional patterns. Harsh conditions imposed by serpentinization (i.e., high pH and low concentrations of terminal electron acceptors and dissolved inorganic carbon [DIC]) presumably exert a strong selective pressure on the associated microbial communities. Our working hypothesis was that to thrive under such challenging conditions, microorganisms of these ecosystems have evolved convergent adaptive strategies that should be observable at the functional level across distant serpentinization sites, beyond apparent differences at the taxonomic level.

Despite the lack of metabolic profile comparisons of alkaline serpentinite-hosted ecosystems so far, a thermodynamic modeling study (16) suggested that peridotite-hosted hydrothermal fields, including LCHF, showed similar catabolic energy profiles available for key microbial metabolisms, while these profiles were distinct from those of basalt-hosted hydrothermal fields. Furthermore, microbiological studies of serpentinizing ecosystems have mainly focused their attention on the most apparent metabolisms associated with the two dominant sources of energy, $H_2$ and $CH_4$, in submarine systems such as LCHF (17) and PBHF (18) and in continental sites such as the Coast Range Ophiolite Microbial Observatory (CROMO; USA) (19), Voltri Massif (Italy) (20), or Santa Elena ophiolite (Costa Rica) (21). However, recent studies have pointed to the potential implication of organic molecules produced by serpentinization and associated reactions, such as formate (22–24) and aromatic amino acids (25), as primary sources of carbon, together with $CH_4$, given the nearly complete depletion of DIC in end member fluids. In contrast, metabolic strategies for nitrogen or sulfur utilization remain

**TABLE 1** Global distribution of studied samples

| Site, country | Type | Sample ID[a] | Depth (m) | pH | Temp (°C) |
|---|---|---|---|---|---|
| Coast Range Ophiolite, USA | Groundwater from well | QV1.1 | −23 | 11.5 | 18 |
| | | CSW1.3 | −23 | 10.1 | 17 |
| | | CSW1.1 | −19 | 12.2 | 17 |
| | | QV1.2 | −15 | 7.9 | 18 |
| Santa Elena Ophiolite, Costa Rica | Spring water | SE.9 | | 11.54 | 26 |
| Voltri Massif, Italy | Spring water | GOR.13 | | 12.3 | 24 |
| | | GOR.12 | | 11.8 | 24 |
| | | BR2.12 | | 12.1 | 22 |
| | | BR2.13 | | 12.3 | 22 |
| Cabeco de Vide, Portugal | Groundwater from a borehole | CVA | −130 | 11.4 | 20 |
| Bay of Prony, New Caledonia | Hydrothermal chimney | PBHF.27 | −43 | 10.6 | <40 |
| | | PBHF.28 | −43 | 10.6 | <40 |
| Lost City, Atlantic Ocean | Hydrothermal chimney | LCHF.75 | −733 | 10.2 | 40–90 |
| | | LCHF.81 | −767 | 10.2–10.7 | 40–90 |
| Axial Seamount, Pacific Ocean | Hydrothermal fluid | Mrk.33 | −1,516 | 5.4 | 28 |
| | | Mrk.113 | −1,522 | 6.2 | 24 |
| Mid-Cayman Rise, Caribbean Sea | Hydrothermal fluid | MCR.SG | −4,940 | 6.7 | 108 |
| Tattapani, Chhattisgarh, India | Water from hot spring | TAT.3 | | 7 | 69 |
| | | TAT.4 | | 7.8 | 67 |
| Octopus Spring, USA | Water from hot spring | Yell.O | | 8 | 84 |
| Conch Spring, USA | | Yell.C | | 8 | 84 |

[a]ID, identifier.

largely understudied in serpentinization-influenced ecosystems (26). Moreover, microbial utilization of P in alkaline serpentinizing systems has not been, to our knowledge, previously addressed, despite P being an essential nutrient for life. The only observation made so far was from prokaryotic lipid biomarkers enriched in the glycosyl head group instead of the phosphorus-containing head group at LCHF, possibly because of low availability of P as a nutrient (27). The most readily assimilated form of P in marine ecosystems is inorganic phosphate (28). However, under phosphate-limited conditions, a wide range of microorganisms can metabolize reduced P compounds as substrates for growth, including phosphonates, which are characterized by a chemically stable carbon-phosphorus (C-P) bond, or phosphite ($PO_3^{3-}$) (29).

In this study, we performed a cross-comparison of metagenomes from submarine and terrestrial serpentinizing systems, along with a selection of metagenomes from other hydrothermal environments, *a priori* not influenced by serpentinization (for example, deep-sea basalt-hosted vents and terrestrial hot springs). We aimed to identify core microbial communities, and common functions or metabolic pathways, which could be characteristic of serpentinizing systems and contribute to understanding of microbial physiology adaptations to these challenging environments.

## RESULTS

**Data set and site location.** The data set consisted of 21 publicly available environmental metagenomes encompassing marine and continental serpentinite-hosted systems, hot springs, and basalt-hosted hydrothermal vents spread worldwide (see Table S1 and Fig. S1 at https://doi.org/10.5281/zenodo.6597409). They originated from 10 well-studied sites comprising PBHF, LCHF, CROMO, Santa Elena ophiolite, Voltri Massif, Cabeço de Vide aquifer (CVA), Tattapani and Yellowstone hot springs, Axial Seamount (Juan de Fuca Ridge), and the Piccard hydrothermal field (mid-Cayman rise). For each metagenome, the type of habitat, the depth, the temperature, and the pH are reported in Table 1. All selected serpentinizing systems emit highly alkaline fluids, with pHs ranging between 9 and 12.3, except for sample QV1.2 in CROMO (pH 7.9). Fluids from other ecosystems had relatively circumneutral pH values.

*De novo* assembly of each metagenome generated between 10,084 and 401,248 contigs, depending on the sample (see Table S2 at https://doi.org/10.5281/zenodo.6597409). This wide variability of contigs resulted from variations in sequencing depth variability

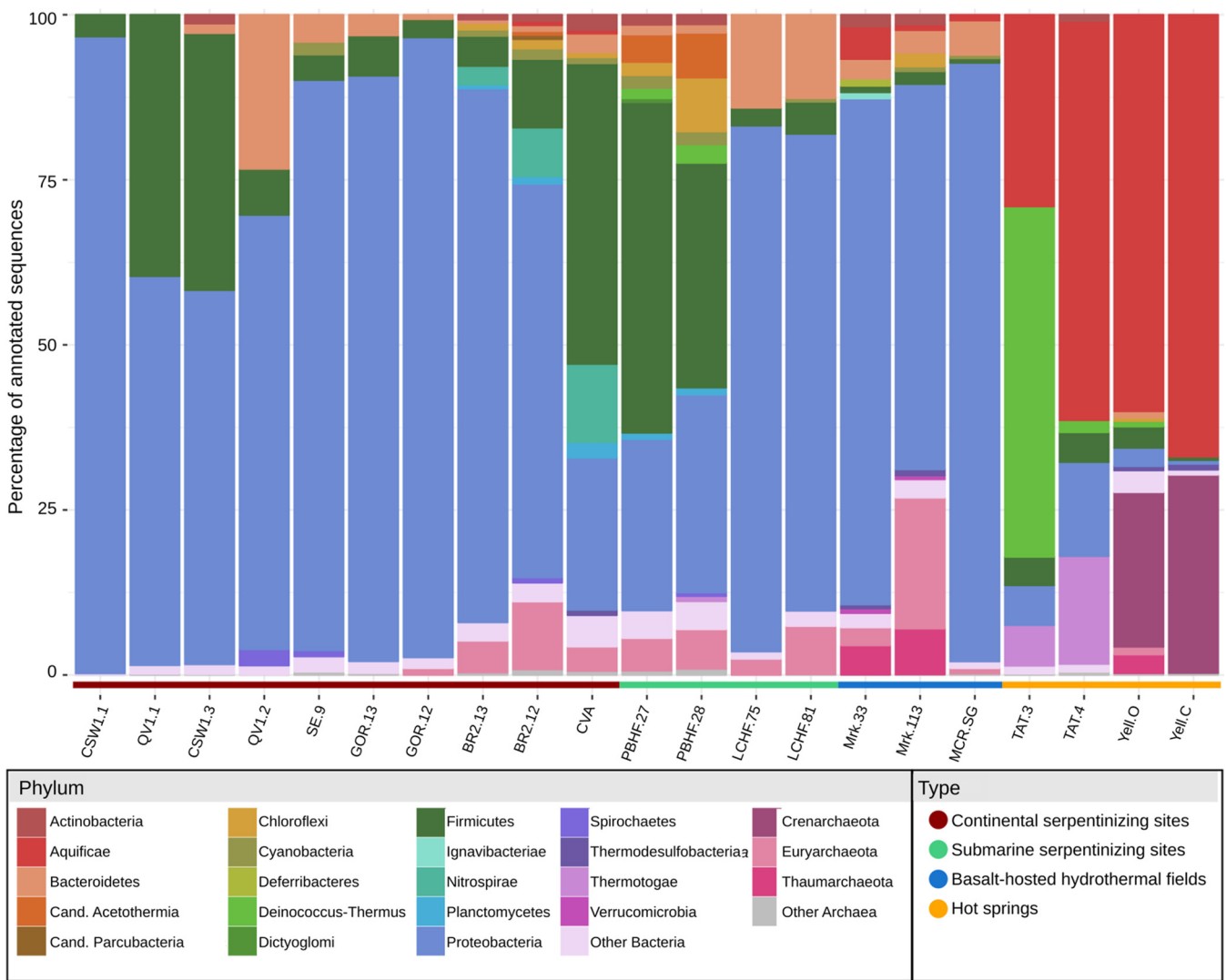

**FIG 1** Relative abundances of microbial phyla in the 21 metagenomes compared in this study. The taxonomic distribution was determined using genes successfully assigned to known prokaryotic taxa with MEGAN6 using the lowest-common-ancestor algorithm.

between metagenomes. Consequently, the number of predicted genes ranged from 16,780 to 579,028. Their rate of annotation to known functional categories reached, on average, 55% and 38% based on the Cluster of Orthologous Groups (COG) and KEGG databases, respectively.

**Microbial community composition.** The overall microbial taxonomic diversity identified in the set of metagenomes was mostly affiliated with 22 phyla (19 bacterial and 3 archaeal phyla [Fig. 1]) based on the taxonomic annotation of all genes. The *Proteobacteria* (up to 96%) and *Firmicutes* (up to 50%) phyla made up the bulk of the microbial composition in most samples. Compared with the other environments, the hot springs showed a distinct taxonomic profile, with a large proportion of thermophilic or hyperthermophilic lineages which is typical of such ecosystems (30). Aside from the hot spring samples, the archaeal communities were dominated by the Euryarchaeota phylum, which represented up to 20% in hydrothermal fields (LCHF, PBHF, and basalt-hosted hydrothermal systems) and continental serpentinizing systems (Voltri Massif and CVA). While being considered marine in this study, the two metagenomes of PBHF harbored taxonomic profiles distinct from those of LCHF or other deep-sea hydrothermal systems. The relatively balanced distribution of the dominant *Firmicutes* and *Proteobacteria* at PBHF was more like that of terrestrial serpentinizing systems (e.g., CVA). However, the abundance of the candidate phylum *Acetothermia* in

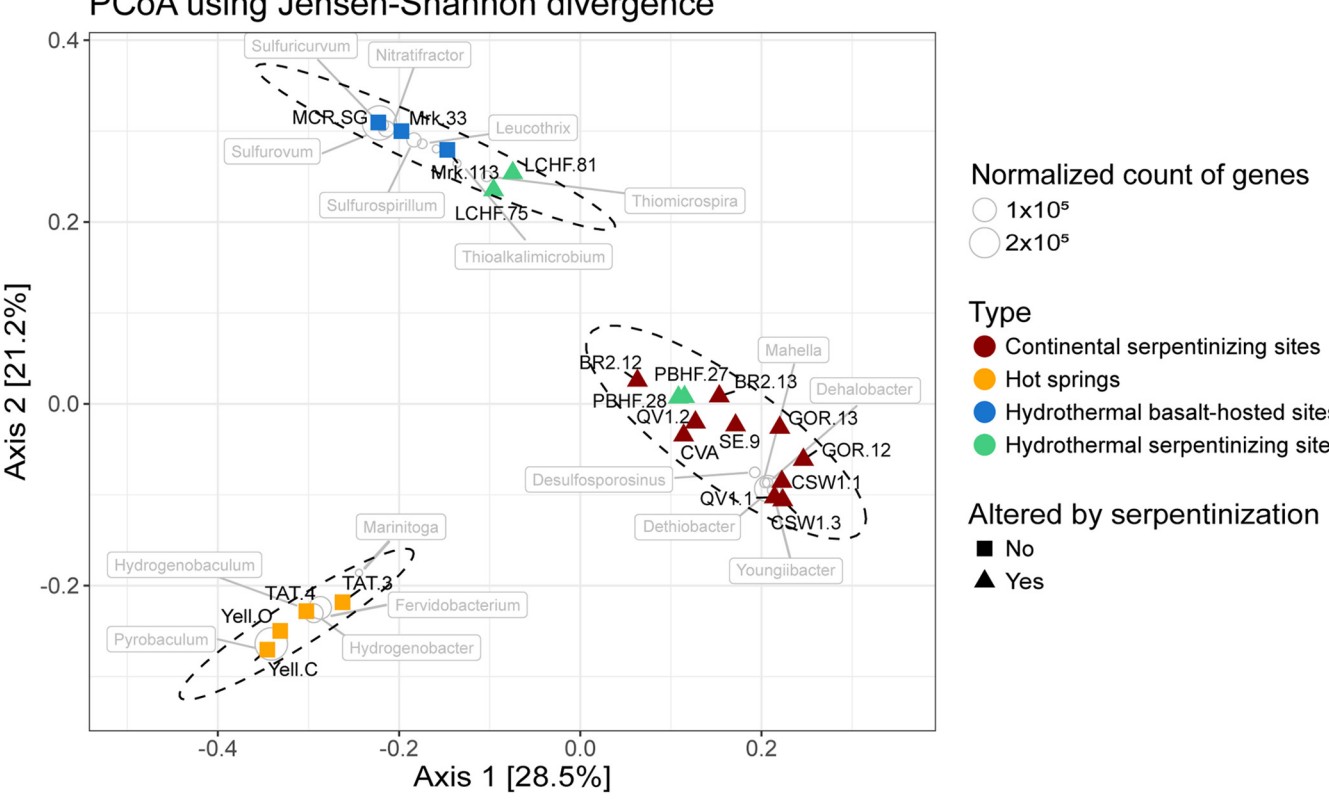

**FIG 2** Microbial community composition at the genus level for the 21 metagenomes. Each metagenome is symbolized by a triangle or a square depending on the presence of serpentinization reactions and is colored by ecosystem category. In each of the 3 clusters, the overabundant genera (with statistically significant differences) are indicated by circles and labeled. The size of circles is proportional to the number of genes assigned to the respective genus.

PBHF constituted a singularity, as the organisms were scarce in other serpentinizing environments compared in this study.

The relationships between metagenomes were further investigated at the taxonomic level of genus, and this analysis revealed a clustering of metagenomes according to the environmental context of the microbiome habitat (Fig. 2). A first cluster gathered the four hot springs, sharing mainly hyperthermophilic genera (e.g., *Hydrogenobacter* and *Pyrobaculum*) typically associated with dihydrogen or iron metabolisms (31, 32). The second cluster included five samples from deep-sea hydrothermal fields, encompassing both basalt and peridotite host rock. Abundant genera found within this cluster included *Gammaproteobacteria* or *Epsilonproteobacteria* mostly associated with sulfur cycling (e.g., *Thiomicrospira* and *Sulfurospirillum* or *Sulfurovum*, respectively) in hydrothermal environments (16). The last cluster contained all the serpentinizing systems fed by fresh water, i.e., continental serpentinizing systems and the coastal PBHF. In this group, the highest abundances were found for members of the *Clostridiales* (e.g., *Youngiibacter* and *Dehalobacter*) and *Betaproteobacteria* (e.g., *Hydrogenophaga*). This analysis thereby set the LCHF metagenomes apart from the ones from all others other serpentinizing systems (Fig. 2).

**Metabolic potential.** The metabolic potential of microbial communities was investigated by searching the metagenomes for genes encoding key enzymes of metabolic pathways involved in utilization of the main products of serpentinization (dihydrogen and methane) or alternative sources of energy (sulfur or nitrogen compounds) or carbon (Fig. 3).

To investigate $H_2$ metabolism, genes encoding [NiFe]- and [FeFe]-hydrogenases were searched in the metagenomes. The bidirectional [NiFe]-hydrogenase-encoding genes of group 3d (33, 34) were more prevalent in all serpentinizing ecosystems (*hoxY* [Fig. 3]) than for other metagenomes. However, this peculiar distribution of gene

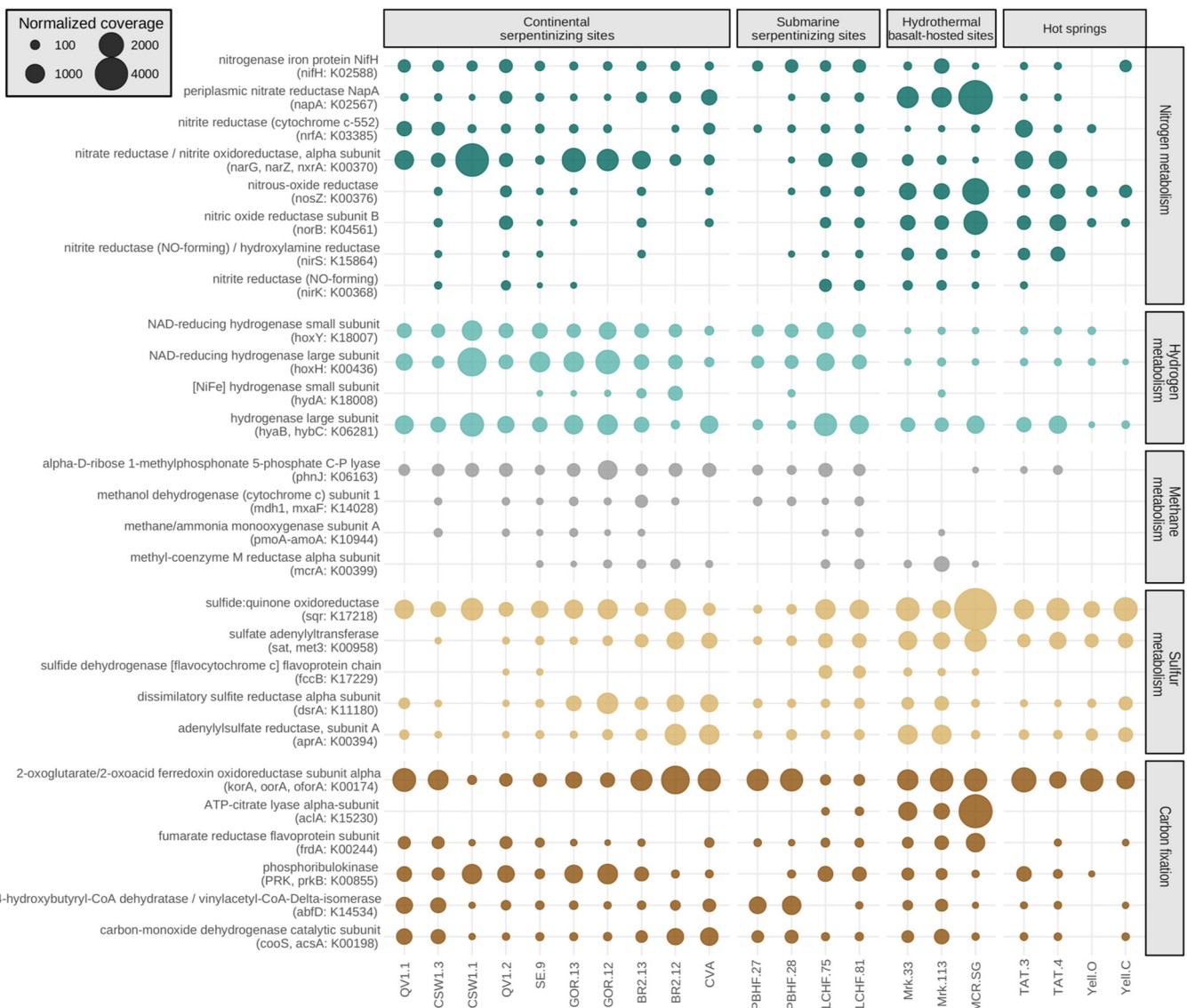

**FIG 3** Potential microbial metabolisms in the 21 metagenomes from hydrothermal systems. Shown is a bubble plot of key genes involved in nitrogen (green), dihydrogen (turquoise), methane (gray), and sulfur (beige) potential metabolisms and carbon fixation (brown). The size of the circles corresponds to the abundance of normalized KOs.

abundance was not observed for other genes associated with $H_2$ metabolism, even those that showed significant differences across sites.

Methanogenesis or anaerobic methane oxidation (AOM) tracked by using the *mcrA* gene (encoding the alpha subunit of the methyl coenzyme M reductase) (35, 36) was detected in half of the serpentinizing sites (not detected at CROMO and PBHF) and in deep-sea vents. Since the presence of *mcrA* in PBHF, which is correlated with the 16S rRNA-encoding gene of *Methanosarcinales*, was consistently reported in previous PCR-based surveys (9, 11), its absence in PBHF metagenomes could be a consequence of their moderate sequencing depth. In contrast, genes involved in aerobic particulate methane oxidation (*pmoA*) or methanol oxidation (*mxaF* or *mdh1*) (37) were detected almost exclusively in serpentinizing systems, which are methane-rich environments.

Both the *nirK* and *norB* genes, selected as genetic markers of denitrification in this study, were detected in only 4 of the 10 continental serpentinizing sites, while they were present in deep-sea vents (including LCHF), and only *norB* was widespread in hot springs. However, key genes of dissimilatory nitrate reduction (representing the first step of denitrification), *napA* and the orthologous *narG*, *narZ*, and *nxrA* genes, were found in all hydrothermal sites.

Although *narG* (and its orthologous genes) was overabundant compared to *napA* in serpentinizing environments (especially in terrestrial sites), the inverse relationship was observed for the deep-sea vents. The *nrfA* gene, a marker of nitrite reduction to ammonium, and the *nifH* gene, a marker for nitrogen fixation, were equally distributed at low abundance in all sites. Finally, genes involved in the anaerobic ammonium oxidation pathway (Anammox), including *hzsA* (encoding the hydrazine synthase) and *hdh* (encoding the hydrazine dehydrogenase), were not detected among the 21 metagenomes.

Regarding sulfur metabolisms, genes associated with the dissimilatory sulfate reduction, such as *dsrA* (dissimilarity sulfite reductase), *aprA* (adenylysulfate reductase), and *sat* (sulfate adenylytransferase), were detected in most of the metagenomes, but with a relatively low abundance at CROMO. In addition, the high abundance of the *sqr* gene (encoding a sulfide:quinone oxidoreductase) suggested that sulfide oxidation is a major metabolism, especially in basalt-hosted hydrothermal vents. Another sulfide-oxidizing metabolism, involving the flavocytochrome *c* sulfide dehydrogenase-encoding gene, was mostly restricted to deep-sea hydrothermal systems, including LCHF.

Most of the key genes of known prokaryotic autotrophic carbon fixation pathways (reviewed by Berg et al. [38]) were recovered in the metagenomes (Fig. 3), except for the *aclA* gene, encoding ATP citrate lyase, one of the key enzymes of the reductive tricarboxylic (rTCA) cycle, which was found only in LCHF and basalt-hosted metagenomes. Other marker genes of autotrophic carbon fixation pathways were differentially represented in the metagenomes of LCHF and PBHF. For example, genetic markers for the 3-hydroxypropionate bicycle (*frdA*) and Calvin-Benson-Bassham cycle ([*prkB*]) were more abundant at LCHF, while at PBHF, the dominant markers were those of the reductive acetyl coenzyme A (acetyl-CoA) (Wood-Ljungdahl) pathways (*cooS* and *acsA*) (Fig. 3).

**Comparison of functional profiles.** Except for *hoxY*, the abundance of targeted metabolic genes did not significantly differ between serpentinizing and other environments. Additionally, the hierarchical clustering of all genes annotated with KEGG Orthology (KO) did not regroup the serpentinizing sites (Fig. S3 at https://doi.org/10.5281/zenodo.6597409) but pointed out functional similarities between LCHF and the basalt-hosted hydrothermal vents. Indeed, their metagenomes were enriched for genes encoding transporters (e.g., for sugar, phosphate, and $Mg^{2+}$) and ion channels as well as for genes often associated with coping strategies under fluctuating environmental conditions, including the *comFC* genes, related to bacterial competence (39), or the *dppA* gene, linked to chemotaxis (40). Likewise, two genes encoding alginate biosynthesis and transport (*alg8* and *eexD*, respectively) were recovered exclusively in metagenomes from deep-sea hydrothermal fields and could contribute to the formation of biofilms (41).

Since the hierarchical clustering of functional profiles did not group serpentinizing sites, a supervised approach was afterwards applied to specifically select annotated genes with a differential abundance between serpentinizing and other hydrothermal systems. It was assumed that the overabundant genes in serpentinizing systems endow beneficial functions to their microbial communities. The random forest method proved efficient in selecting a set of genes whose profile segregated the serpentinizing sites from other hydrothermal environments (Fig. 4). The genes specifically enriched in serpentinizing ecosystems belonged to various functional categories, including cellular responses to exposure to stressful conditions (heat shock response [*hcrA* and *rpoE*]) as well as DNA repair (*recO*) and defense mechanisms (*fitB*, *ndoAI*, and *cas7*). Genes encoding membrane transporters of antibacterial compounds microcin C (*yejA*) and the ABC-type $Fe^{3+}$-siderophore transport system (ABC.FEV.A) were also overabundant in serpentinizing metagenomes. This is also the case for the gene (*mnhD*) encoding part of a putative multisubunit $Na^+/H^+$ antiporter (Mnh complex), involved in homeostasis of the cytosolic pH, and membrane energization (42), which is expected to be important for life in hyperalkaline environments. As mentioned above, the metagenomes from serpentinizing systems were enriched in *hoxH* and *hoxY*, which encode the large and small subunits of an [NiFe]-hydrogenase of group 3d. However, one of the most striking features was the

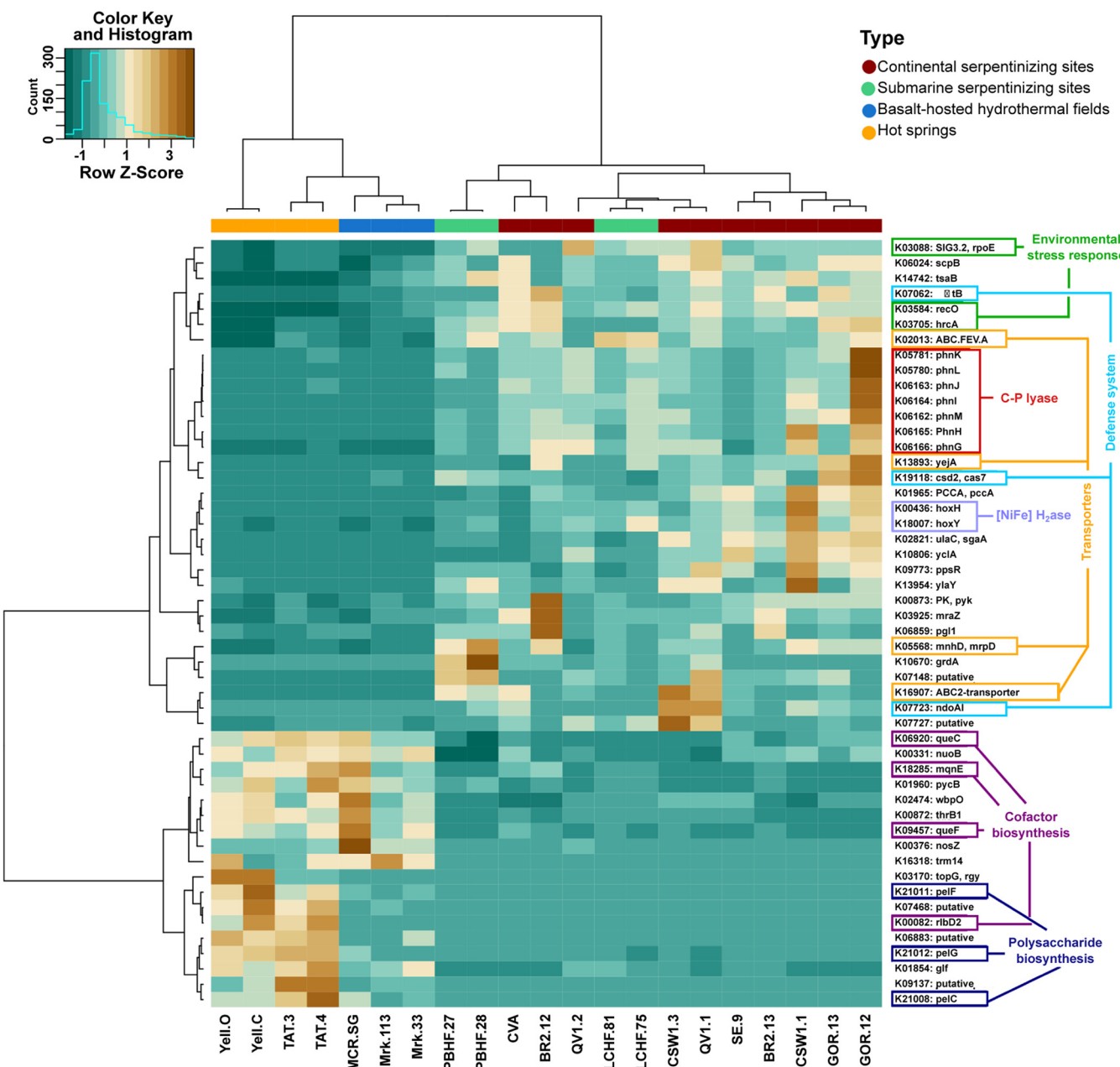

**FIG 4** Heat map showing the normalized abundance of the top 50 KEGG Orthology annotations that best discriminate the 21 functional profiles into two categories of ecosystems. The brown tint indicates a higher relative abundance.

overrepresentation of seven genes from the *phn* operon, encoding enzymes for phosphonate catabolism. The same overrepresentation of *phn* genes was obtained using a COG-based annotation in the random forest clustering (Fig. S4 at https://doi.org/10.5281/zenodo.6597409).

**The carbon-phosphorus lyase operon.** The seven identified genes (i.e., *phnGHIJKLM*) belong to the operon coding for the C-P lyase and, therefore, are involved in the breakdown of various phosphonates (43). These genes constitute the minimal catalytic core unit, essential for C-P bond cleavage (44). Remarkably, the seven genes (*phnGHIJKLM*) of the C-P lyase were almost exclusively present in serpentinite-hosted ecosystems (Fig. 5 and Table S3 at https://doi.org/10.5281/zenodo.6597409). The even abundance of *phn* genes within each metagenome most likely reflected their organization in an operon. In the serpentinizing sites studied, we estimated that 5% to 52% of microbial genomes (according to the normalization against the abundances of 10 single-copy housekeeping genes) could

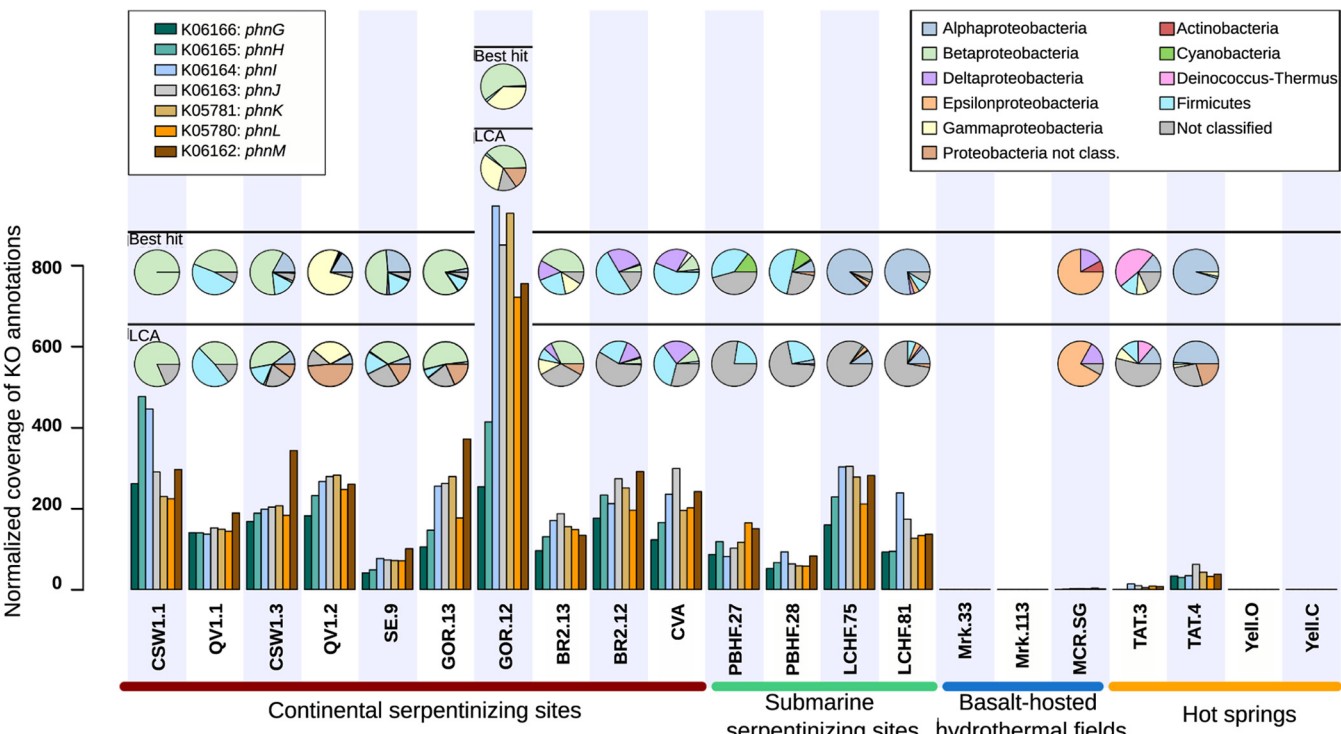

**FIG 5** Abundances of genes encoding the catalytic component of C-P lyase in the 21 metagenomes investigated in the present study. The bar plot corresponds to the normalized distribution of each KO annotation associated with *phnGHIJKLM* genes. Pie charts indicate their taxonomic distribution per metagenome among phyla and proteobacterial classes.

contain the C-P lyase essential genes, a rate comparable to the 30% detected in the metagenomes of the Sargasso Sea, where phosphonate degradation is a prominent process (45) (see Table S4 at https://doi.org/10.5281/zenodo.6597409). Taxonomic annotation of the *phn* genes identified in the metagenomes distributed them in several microbial phyla (Fig. 5). They were predominantly affiliated with *Alphaproteobacteria* and *Firmicutes* at LCHF and PBHF, respectively. In the continental serpentinizing ecosystems, most of *phn* genes were affiliated with *Betaproteobacteria* members. Phylogenetic inferences of the *phnJ* gene, encoding the catalytic subunit involved in the cleavage of the C-P bond, were used to ascertain these assignments (Fig. S6 at https://doi.org/10.5281/zenodo.6597409). However, a high percentage of *phn* genes could not be assigned to known taxa ("not classified" in Fig. 5) with the lowest-common-ancestor (LCA) algorithm, especially in submarine sites.

In our metagenomic data set, the genomic organization of the *phn* operon differed from the canonical form (*phnGHIJKLM*) by gene order rearrangements and frequent insertion of additional genes encoding acetyltransferases, transporters, or phosphatases (Fig. S5 at https://doi.org/10.5281/zenodo.6597409). Such rearrangements are relatively common, as well as is the integration of other genes related to phosphorus cycling (44). In particular, *phn* gene operons reconstructed from the metagenomes of our study included genes coding for phosphonate transporters (*phnCDE*). Although the phosphonate transporter-encoding genes seemed particularly enriched in serpentinizing ecosystems than for other metagenomes (except for TAT.3 [Table S3 at https://doi.org/10.5281/zenodo.6597409]), we observed a notably low detection of genes involved in different phosphonate biosynthesis pathways. In some metagenomic *phn* operons, the *phnF* gene (encoding a transcriptional regulator) is replaced by other genes (*dasR*, *yvoA*, or *yycF*) encoding transcription regulators of the GntR family normally implicated in the regulation of genes from the *N*-acetylglucosamine-degrading pathway (46). Finally, an atypical operon identified in the metagenomes of PBHF showed an insertion of a NAD-phosphite oxidoreductase gene (*ptxD*, encoding phosphite degradation) between the

genes *phnE* and *phnG* (Fig. S5 at https://doi.org/10.5281/zenodo.6597409). This novel genomic organization could degrade phosphonate or phosphite since the Pn transporter encoded by *phnCDE* genes can also transport the phosphite ion inside the cells (29).

Compared to the case for the C-P lyase pathway, the genes involved in alternative phosphonate degradation pathways (47) were more sparse in serpentinizing systems, in particular genes involved in 2-aminoethylphosphonate degradation (reported in Table S3 at https://doi.org/10.5281/zenodo.6597409). Although we noted that genes involved in phosphate transport (*pstSABC*) were widely distributed in all metagenomes, the stress marker gene (*phoB*) for the inorganic phosphorus (Pi) depletion regulatory system (48) was enriched in serpentinite-hosted environments (in addition to TAT.3), but not in PBHF and CVA (Table S3). Such trends were not observed to serpentinization associated metagenomes for other genes potentially involved in Pi depletion, including the *lpxH*-like gene for cell membrane phospholipids substitution. Contrasting with genes involved in phosphonate catabolism, those encoding enzymes involved in phosphonate biosynthesis were almost only detected in deep-sea hydrothermal field (Tables S3 and S5).

## DISCUSSION

**Microbial communities in serpentinizing submarine ecosystems.** Our analysis based on the taxonomic profiles showed that PBHF microbiomes clustered with those of continental serpentinizing systems, while LCHF microbiomes grouped with the basalt-hosted deep-sea vents. Although this result could be a consequence of sampling strategies (23, 24) or sequencing depth, it was unexpected since numerous phylotypes previously identified as specific to serpentinizing systems (13) including the LCMS (Lost City Methanosarcinales) or members of the genus *Hydrogenophaga* (9, 13) were indeed retrieved exclusively from the serpentinite-hosted metagenomes. Here, the peculiar situation of LCHF could be explained by the sequences assigned to sulfur oxidizers (e.g., *Gammaproteobacteria* and *Epsilonproteobacteria*) highly abundant in both LCHF and basalt-hosted deep-sea vents but nearly absent in all others serpentinizing sites (Fig. 2). Sulfur-oxidizing bacteria are typically found in sulfidic environments such as deep-sea hydrothermal vents (16, 49, 50), where they thrive by oxidation of reduced sulfur compounds at low oxygen tensions. Consistent with this observation, sulfides are barely detectable in continental serpentinite springs and PBHF, whereas the $H_2S$ concentration in LCHF is more in the range (0.2 to 3 mM) of that of basalt-hosted hydrothermal vents, likely owing to its setting proximal to the Mid-Atlantic Ridge and its black smokers (51, 52). However, recent studies demonstrated the key role of sulfur-cycling in supporting microbial life in continental serpentinite-hosted systems, while potentially driven by phyla other than *Epsilonproteobacteria* or *Betaproteobacteria* (53, 54).

**Functional capabilities within serpentinizing ecosystems.** The overabundance of the genes encoding [NiFe]-hydrogenase group 3d (*hoxHY*) in all serpentinizing sites implies an importance of $H_2$ metabolism across terrestrial and submarine systems. This cytoplasmic hydrogenase was presumably essential for autotrophic growth of members of the *Betaproteobacteria* in The Cedars and CROMO serpentinites springs (19, 55). Such hydrogenases are known to be oxygen tolerant in *Ralstonia eutropha* H16 (56) uptake or fermentative production depending on electron acceptor availability (33, 34, 57, 58), which may be of benefit to microbial communities living at the interface between anoxic and oxic zones.

No specificity related to nitrogen metabolism was observed for microbial communities of serpentinizing systems compared to other hydrothermal systems. The diversity of geochemical characteristics in the serpentinizing systems made the neutral distribution of the *nifH* gene surprising. Indeed, nitrogen fixation was previously assumed to be more favorable under highly reducing and energetic conditions of submarine ecosystems rather than in continental environments (59). Further, dissimilatory nitrate reduction can also occur in almost all serpentinizing sites studied, based on the distribution of genetic markers. This pathway is presumed to be more advantageous than denitrification in marine environments,

because ammonia can be reused as a nitrogen source without expending further energy toward nitrogen fixation (60).

Despite the high concentration of methane in serpentinizing sites, the potential for methanotrophy or methanogenesis differed from one site to another (Fig. 3). Biological methane production has been demonstrated in some serpentinizing ecosystems but is inconsistent between sites (20, 21, 61). The variable origins of methane within these sites probably result from distinct mixtures of thermogenic, abiogenic, and microbial methane as proposed for two serpentinizing systems, CROMO and The Cedars (62). At CROMO, although known pathways for methanogenesis were not reported, a *Betaproteobacteria* and *Clostridia* were positively correlated with the concentration of methane (19). Here, these two taxonomic groups were abundantly identified in the terrestrial serpentinizing systems and shown to be linked to C-P lyase, hence nurturing the hypothesis that phosphonate catabolism is active and leads to methane production.

Unexpectedly, the extreme conditions of life imposed by serpentinization reactions did not lead to a convergence of the related functional profiles of the investigated genes (see Fig. S3 at https://doi.org/10.5281/zenodo.6597409). However, the similarities between LCHF and other deep-sea hydrothermal systems highlighted by our hierarchical clustering have already been reported through enrichments of genes involved in stress responses, homologous recombination, and chemotaxis (63). Microbial communities in these systems are subjected to numerous physical constraints (e.g., hydrostatic pressure and temperature) and chemical stresses (e.g., heavy metals and radionuclides) that are probably specific to deep-sea hydrothermal vents and reflected in their genomic adaptations. Additionally, high abundances of mobile genetic elements such as transposases were also reported as a distinctive characteristic of LCHF and deep-sea hydrothermal vents (64, 65). It has been suggested that lateral gene transfer is frequent and could be an important source of phylogenetic diversity (65) as well as a way to promote bacterial adaptation to ecological niches. In our study, the high abundance of mobile genetic elements did not appear to be restricted to deep-sea hydrothermal vents, as they were also very common in other systems (data not shown).

**Metabolism of reduced phosphorus compounds in serpentinizing systems.** Phosphate ($PO_4^{3-}$), commonly used as a source of phosphorus (P) in aquatic ecosystems, was reported at very low concentrations ($<1$ $\mu$M) in several geochemical studies of serpentinizing systems (66), raising questions about the source of P and its mechanism of assimilation in serpentinite-hosted ecosystems. Indeed, phosphorus is an essential nutrient owing to its pivotal role in cell structure, storage, metabolism, and gene expression for all living organisms. The supervised approach highlighted the importance of the genes *phnGHIJKLM* in serpentinizing systems. These genes, encoding the functional core of the C-P lyase, should enable microorganisms to utilize phosphonates as source of P. In phosphate-limited marine environments, phosphonates are often considered an important source of P and could be metabolized by diverse microbial taxa, notably among *Proteobacteria* (45, 48). Among the enzymes involved in phosphonate catabolism, the C-P lyase pathway has been extensively studied because its enzymatic activity acting upon methylphosphonate esters as the substrate has been proposed as an explanation for the biological production of methane in oxygenated environments (48, 67, 68). Considering the abundance of genes involved in this pathway in serpentinizing systems (Table S3 at https://doi.org/10.5281/zenodo.6597409), as well as the enrichment of its regulon-encoding gene (*phoB* [Table S3]), we hypothesize that phosphonate catabolism could play an important role in the growth of microbial communities. The cleavage and incorporation of phosphonates in serpentinizing systems could prove an important means of phosphorus scavenging in this low-nutrient environment. Further, it is likely that in serpentinizing systems, methane or other hydrocarbons could be partially produced during the microbial degradation of phosphonates to inorganic phosphate (Fig. 6). The type of phosphonate substrates remains to be to identified, because the C-P lyase cleaves a broad range of substrates according to the species, including alkylphosphonates, 2-aminoethylphosphonate, and phenylphosphonate (69).

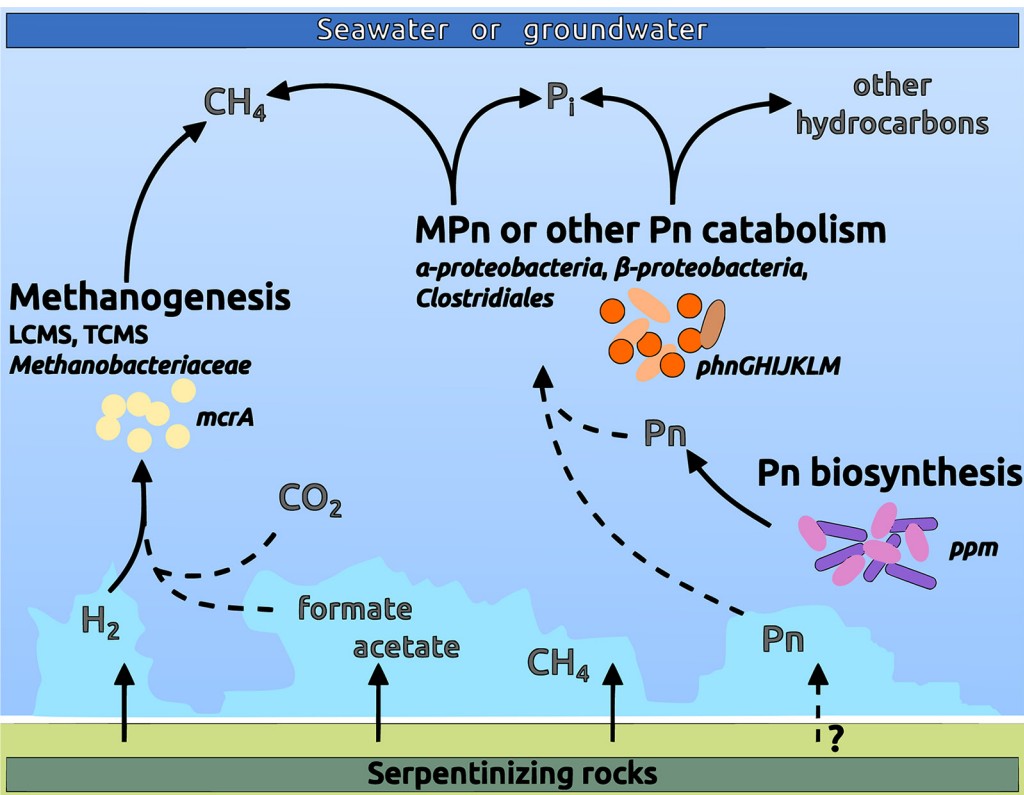

**FIG 6** Schematic of the proposed links between the carbon and phosphorus cycles in serpentinizing environments. Reactions whose origin of the chemical compounds is unknown are indicated by dashed lines.

While other pathways are known to cleave C-P bonds, they require very substrate-specific hydrolases, such as phosphonoacetaldehyde hydrolase, phosphonoacetate hydrolase, and phosphonopyruvate hydrolase (70), or an oxidative cleavage with the 2-amino-1-hydroxyethylphosphonate dioxygenase (71), for which respective encoding genes showed a lower abundance, when detected, in serpentinite-hosted ecosystems than all genes involved in the C-P lyase pathway. The presence of many alternative pathways in the Santa Elena Ophiolite is therefore surprising, as the phosphate concentration of this site was higher than in other serpentinizing environments (21, 72). This result may be explained by the observation that the catabolism of phosphonates was not restricted to the phosphate-limited environment, being inorganic phosphate insensitive (73). This hypothesis is also supported by the occurrence of 3 Pi-independent phosphonate transporter-encoding genes (*aepX*) in its metagenome (74).

Although the C-P lyase pathway is now recognized as widely distributed in the contemporary biosphere (75), its identification in ecosystems that may have supported early life suggests a potential link to ancient metabolisms. Phosphonates were assumed to be present on the early Earth (76) because alkylphosphonates were detected in the Murchison meteorite (77). Furthermore, several mechanisms for organophosphate synthesis under prebiotic conditions have been reported (78) and may contribute to the tapestry of geochemical processes occurring in modern-day serpentinizing environments. At present, geochemical data are missing to identify the origins of phosphonates in serpentinizing sites. Alternatively, the detection of a phosphonate synthesis gene (*pepM*) could be associated with the presence of phosphonate in dissolved organic matter (68) and be a path to understand its distribution. However, genetic markers (*pepM*, *ppd*, *phpC*, and *mpnS*) (79) of this biosynthesis pathway were rarely retrieved in the metagenomes in this study (see Tables S3 and S5 at https://doi.org/10.5281/zenodo.6597409). Furthermore, abundances of these genes were not in concordance with those of C-P lyase-encoding genes. In most marine ecosystems, phosphonate could be synthesized by

some phylotypes of *Cyanobacteria* and Thaumarchaeota (80), two groups also usually found in serpentinizing ecosystems. Nonetheless, their ecological importance was not discussed here due to their low abundance, despite their potential strong adaptation to this extreme environment (81). Therefore, we could not identify any microbial phosphonate producer in these metagenomes and, thus, a biological source of phosphonates. Nonetheless, such sources cannot be excluded and should be explored in detail in future studies. Among the plausible hypotheses on the origin of phosphonates in serpentinized ecosystems, we propose (i) an external source that implies the transport of phosphonates to these ecosystems or (ii) a biological synthesis of phosphonates by an unknown metabolic pathway not dependent on *pepM*. In the first scenario, microbiomes influenced by serpentinization would benefit from the input of organic P sources which may be derived from fluids feeding the ecosystem (e.g., meteoric fluids or seawater) or stored in substratum rocks and released during alteration. Indeed, phosphonate, like phosphate, could be absorbed to metal (oxy)hydroxides (82) or carbonates, two secondary minerals produced by serpentinization (83). In Pi-limited marine settings, some heterotrophic bacteria are known to secrete metal chelators or redox-active antibiotics to dissolve these iron minerals to release P compounds (81, 84). Although we cannot conclude that such metabolisms are feasible in serpentinizing ecosystems, we observed an enrichment of genes coding for antibacterial (*yejA*) and iron complex (ABC.FEV.A) transporters together with C-P lyase pathway genes in their relative metagenomes (Fig. 5). Altogether, these results point toward the need to resolve sources of phosphonates that could contribute to the occurrence of this pathway.

**Conclusion.** This first comparative metagenomic study of serpentinite-hosted environments, although limited by the coherence of sampling strategies and the size of the data set, provides an objective framework to understand the functioning of these peculiar ecosystems. We showed a taxonomic similarity between PBHF and other terrestrial serpentinite-hosted ecosystems. At the same time, the LCHF microbial community was closer to deep basalt-hosted hydrothermal fields than continental ophiolites, despite the influence of serpentinization.

This study revealed shared functional capabilities among serpentinite-hosted ecosystems in response to environmental stress, the metabolism of abundant dihydrogen, and the metabolism of phosphorus for the first time in such environments. Our results are consistent with the generalized view of serpentinite environments but provide deeper insight into the array of factors that may control microbial activities in these ecosystems. Moreover, we show that metabolism of phosphonate, through the C-P lyase pathway, is widespread among alkaline serpentinizing systems and could play a crucial role in phosphorus and methane biogeochemical cycles. This study opens a new line of investigation of the metabolism of reduced phosphorus compounds in serpentinizing environments combining metagenomics and activity-based studies. From a broader perspective, similar approaches including more comprehensive multiomics data from serpentinite-hosted microbiomes could better constrain the microbial ecology of such ecosystems.

## MATERIALS AND METHODS

**Metagenomic data sets.** Twenty-one publicly available metagenomic data sets from 6 serpentinizing systems (2 submarine, 4 terrestrial, and 4 other hydrothermal systems [2 deep-sea basaltic hydrothermal vents and 2 hot springs from continental volcanic area]) were retrieved from the NCBI Sequence Read Archive (SRA) and MG-RAST databases and are described in Table S1 at https://doi.org/10.5281/zenodo.6597409. To limit the bias arising from the use of different sequencing technologies and protocols, we selected only metagenomic data sets containing Illumina raw paired-end reads with a partial overlap.

**Comparative metagenomic pipeline.** With the objective of standardizing data processing, all 21 metagenomes (see Table S1 at https://doi.org/10.5281/zenodo.6597409) were reprocessed using an in-house bioinformatic pipeline for quality control, assembly, and annotation against taxonomic and functional databases. This pipeline was implemented in Snakemake (85) and deposited on the Github repository at https://github.com/elfrouin/MetaGPipeline.git. The raw paired-end reads of metagenomes were first merged using PandaSeq v2.8 (86) with default parameters. The merged reads were next processed by Trimommatic v0.32 (87), which trimmed reads once the average quality in a 4-bp window fell below 15, cut bases off the extremities if below a quality score of 20, and removed short reads (<35 bp). The high-quality trimmed reads were finally assembled into contigs, longer than 200 bp, with IDBA-UD v1.1.1 (88). The quality of assemblies was checked by estimating the rate of

reads mapped to the contigs with BWA v0.7.12 (89). Gene prediction was done with Prodigal v47 (90), using the "-p meta" flag for metagenomic sequences. Gene coverage was computed with Bedtools v2.25.0 (91). The microbial taxonomic diversity of each metagenome was determined from the taxonomic annotation of predicted genes. Similarity searches were first performed against the NCBI nonredundant (nr) database with DIAMOND BLASTP v0.8.34 (92) using a maximal E value of $1 \times 10^{-5}$. MEGAN6 (93) was used to assign the annotated genes to their most confidently predicted taxonomic rank, by applying the lowest-common-ancestor (LCA) algorithm. A taxonomic contingency table was created by weighting annotated genes by their sequencing coverage. A first functional annotation of genes was performed by RPS-BLAST search (E value cutoff of $1 \times 10^{-5}$) against the Cluster of Orthologous Groups (COG) database (94). The second functional annotation was performed using the online tool GhostKOALA (the only nonautomated step) against the KEGG Orthology (KO) database with a minimum alignment score threshold of 50 (95). The coverage of functional genes was normalized between metagenomes using a scaling factor estimated by the trimmed mean of M values (TMM) method (96), implemented in the R package edgeR v3.14.0 (97). This normalization method was proposed to be more accurate for shotgun metagenomic analysis, in particular for the detection of differentially abundant genes (98, 99).

**Statistical analyses.** The taxonomic similarity between the 21 metagenomes was determined using relative abundances of archaeal and bacterial genera. Based on the normalized contingency table (exported from MEGAN6), a distance matrix was computed with the Jensen-Shannon divergence using the R package phyloseq v1.20.0 (100). The results were visualized using principal-coordinate analysis (PcoA) (Fig. 2). A nonparametric Kruskal-Wallis test combined with pairwise Wilcoxon tests (adjusted using Bonferroni's correction) was performed to detect genera with significant differential abundance among the subsets of metagenomes identified by PCoA. These specific microbial genera were added on the PCoA biplot.

Regarding the functional capabilities of the microbial community, the Pearson correlation and Ward's method were employed for agglomerative hierarchical clustering of metagenomes and annotated genes. The gene coverage per metagenome was visualized using heat maps, generated with the R package gplots v3.0.1. To discriminate the functional capabilities of serpentinizing versus nonserpentinizing systems, a random forest analysis was carried out using the R package Random Forest v4.6.12 (101) with 3,000 independent decision trees. Finally, a heat map was built with the top 50 genes contributing to segregate metagenomes related to serpentinizing ecosystems from the others (i.e., the genes with the highest value of mean decrease Gini, a measure of variable importance).

**phn operand analysis.** Analyses of *phn* genes were carried out to identify their associated taxonomies and genomic organizations. Genes were queried against the NCBI nr database using BLASTN (with an E value of $1 \times 10^{-6}$). The determination of gene taxonomy was next assessed in two ways: (i) via the best BLAST hit and (ii) via the LCA algorithm implemented in MEGAN6. To estimate the percentage of microorganisms possessing a *phn* operon, the occurrences of *phnGHIJKLM* genes were normalized by the median abundance of 10 single-copy genes (listed in Table S6 at https://doi.org/10.5281/zenodo.6597409) in each sample. Finally, the structural organization of the operon *phn* was investigated from the metagenomic contigs. All contigs with at least the seven genes of the catalytic unit were reannotated with PROKKA v1.11 (102) to characterize the genomic context of the operon.

**Phosphonate transport and biosynthesis.** Despite the abundance of C-P lyase-encoding genes in our data set, well-known genes involved in phosphonate biosynthesis (i.e., *pepM*, *ppdA*, *ppdH*, and *mpnS*) were poorly detected with the approach described above. Therefore, we further examined the occurrence of these genes to confirm our results. For this purpose, we searched for homology against collections of hidden Markov models (HMMs) using the *hmmsearch* tool implemented in HMMER v3.3.2 (hmmer.org). First, homologs of genes were searched against the HMM database proposed by Acker and colleagues (80) and publicly available in their GitHub repository (https://github.com/slhogle/phosphonates/tree/master/data/phosphonate_biosynthesis/HMM_models). Furthermore, we looked for the presence of the highly conserved active-site motif EDK(X)5NS in sequences identified as PepM, since HMMs of PepM present significant similarities with the isocitrate lyase superfamily (78).

Additionally, we carried out a similar approach on the recently described *aepX* gene (74), encoding a Pi-independent phosphonate transport, using a curated database. Briefly, homologs of the *aepX* gene were identified using the Integrated Microbial Genomes (https://img.jgi.doe.gov/) homolog display tool and aligned with ClustalOmega implemented in the IMG/JGI website. Then, aligned amino acid sequences were profiled with the *hmmbuild* function. Sequence homology was searched in each metagenome applying the *hmmsearch* function with an E value of $1 \times 10^{-80}$.

## ACKNOWLEDGMENTS

This project was financially supported by ANR projects deepOASES (14-CE01-0008-06) and MICROPRONY (19-CE02-0020-02), the Deep Carbon Observatory (Census for Deep Life: Comparative metagenomics of archaeal biofilms in carbonate chimneys associated with geographically isolated sites of serpentinization; principal investigator, M. O. Schrenk; funded by the Alfred P. Sloan Foundation). E.F. was awarded a Ph.D. fellowship from the French Ministry of Education and Scientific Research.

E.F., F.A., and G.E. designed the study. E.F., A.L., and F.A. performed the bioinformatics analyses and interpretations. E.F., A.L., F.A., and G.E. wrote the manuscript. M.O.S. reviewed drafts of the paper.

We declare no competing interests.

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
