## [Reviewer comments · mSystems]

Comparative metagenomics highlight a widespread pathway involved in the catabolism of phosphonates in marine and terrestrial serpentinizing ecosystems

Eléonore Frouin, Aurélien Lecoivre, Fabrice ARMOUGOUM, Matthew Schrenk, and Gaël ERAUSO

Corresponding Author(s): Gaël ERAUSO, Mediterranean Institute of Oceanography

Review Timeline:

Submission Date:	April 13, 2022
Editorial Decision:	May 4, 2022
Revision Received:	June 14, 2022
Accepted:	June 15, 2022

Editor: Jack Gilbert

Reviewer(s): The reviewers have opted to remain anonymous.

Transaction Report:

DOI: <https://doi.org/10.1128/msystems.00328-22>

May 4, 2022

Prof. Gaël ERAUSO
Mediterranean Institute of Oceanography
Marseille
France

Re: mSystems00328-22 (Comparative metagenomics highlight a widespread pathway involved in the catabolism of phosphonates in marine and terrestrial serpentinizing ecosystems)

Dear Prof. Gaël ERAUSO:

Thank you for submitting your manuscript to mSystems. We have completed our review and I am pleased to inform you that, in principle, we expect to accept it for publication in mSystems. However, acceptance will not be final until you have adequately addressed the reviewer comments.

Preparing Revision Guidelines

Sincerely,

Jack Gilbert

Editor, mSystems

Journals Department
Reviewer comments:

Reviewer #1 (Comments for the Author):

I'd like to thank the authors for the work they've done to the manuscript since its first submission, and I think the manuscript is in much better shape now. Overall I think that the authors have addressed the major concerns I had with the original manuscript.

I have a few small suggestions for the revised manuscript:

- Line 110, "The most readily assimilated form of P in marine ecosystems is the inorganic phosphate" - "is the inorganic phosphate" could simply be "is inorganic phosphate" here.
- Line 185: "Although narG (and its orthologous genes) was overabundant compared to napA in serpentinizing environments (especially in terrestrial sites), the inverse relationship was observed for the deep-sea vents and hot springs." - This does appear to be true for deep-sea/hydrothermal basalt-hosted sites, but narG is more abundant in the TAT3/4 hot spring sites. The other two hot spring sites appear to be lacking reads for these genes, so perhaps the statement about the inverse relationship should only be applied to the deep-sea sites.
- Line 388, "[phosphonate] biosynthesis pathway were rarely retrieved in the studied metagenomes (Fig. 6; Tables S3 and S5)" - I don't think the link to Fig. 6 makes sense here.
- Line 396 - "which may derived from fluids feeding the ecosystem" should read "which may be derived from fluids feeding the ecosystem".
- Figure legend 3: "Bubble plot of key genes involved inmethane (blue)," - Perhaps this is my computer, but those bubbles look grey to me rather than blue.
- Table S3: Do the authors have data for the prevalence of the phnA gene? As this is another common 2-AEP-specific degradation pathway.
- Table S5 legend: "Occurrences of phosphonates tranport" should read "Occurrences of phosphonate transport"

Reviewer #2 (Comments for the Author):

Most responses to both reviewers have improved the manuscript. However, there are still a couple of minor concerns listed, below. Once these have been dealt with, the manuscript will be acceptable for publication.

First, Reviewer 1 has asked for more discussion on the apparent sources of phosphonates in these environments. I do not believe the authors have specifically addressed this comment. Indeed, there is no direct response in the rebuttal. I see the authors have amended the discussion (Line 393) and said that Acker et al. found a negative correlation between producers and degraders, which is an incorrect statement. Acker et al found the degraders and consumers represented distinct cells, and these traits were rarely found in the same organisms. I have reread this manuscript and I can't see any data or discussion on the abundance of production versus consumption genes within the metagenomes, specifically. The only data I see if figure 3A & B, that show number of SAGs in the Pacific possessing either genes sets is positively correlated with depth. Therefore, I do not believe that the authors have adequately answered reviewer 1's comment. Sorry.

Line 272 - This sentence mentions alternative degradation pathways, but no references are given. The authors should include this review (see below) as a minimum, otherwise readers will not really know what genes are being mentioned, without downloading and screening the supp data. In addition, a non-specialist will not know so much about those genes.

Villarreal-Chiu, Juan F et al. "The genes and enzymes of phosphonate metabolism by bacteria, and their distribution in the marine environment." *Frontiers in microbiology* vol. 3 19. 26 Jan. 2012, doi:10.3389/fmicb.2012.00019

Line 376 and 501 - the authors mention AepX and phosphate-insensitive mineralisation and don't add the appropriate references. Indeed 501 (methods), it seems like the authors forgot to add as the have empty parentheses.

Can you please add the coverage for aepX to table S3, so there is a direct comparison. In table S5, the number of genes identified is quite similar to phnD in 7/14 sites. The read coverage would presumably show a better difference and add weight to the idea that C-P lyase is more widespread and specific to these systems. You also do not have PhnA (hydrolase), which had something in the previous version. In most marine systems PhnWAY is used by AepX producing marine bacteria, not PhnWX. Adding this would really help clear up this concern.

June 15, 2022

Prof. Gaël ERAUSO
Mediterranean Institute of Oceanography
Marseille
France

Re: mSystems00328-22R1 (Comparative metagenomics highlight a widespread pathway involved in the catabolism of phosphonates in marine and terrestrial serpentinizing ecosystems)

Dear Prof. ERAUSO:

Your manuscript has been accepted, and I am forwarding it to the ASM Journals Department for publication. For your reference, ASM Journals' address is given below. Before it can be scheduled for publication, your manuscript will be checked by the mSystems production staff to make sure that all elements meet the technical requirements for publication. They will contact you if anything needs to be revised before copyediting and production can begin. Otherwise, you will be notified when your proofs are ready to be viewed.

Publication Fees:

We recognize that the video files can become quite large, and so to avoid quality loss ASM suggests sending the video file via <https://www.wetransfer.com/>. When you have a final version of the video and the still ready to share, please send it to mSystems staff at mSystems@asmusa.org.

For mSystems research articles, if you would like to submit an image for consideration as the Featured Image for an issue, please contact mSystems staff at mSystems@asmusa.org.

Sincerely,

Jack Gilbert
Editor, mSystems

Journals Department
E-mail: mSystems@asmusa.org